# All-Atom Molecular Dynamics Simulations of the Temperature Response of Poly(glycidyl ether)s with Oligooxyethylene Side Chains Terminated with Alkyl Groups

**DOI:** 10.3390/nano13101628

**Published:** 2023-05-12

**Authors:** Erika Terada, Takuya Isono, Toshifumi Satoh, Takuya Yamamoto, Toyoji Kakuchi, Shinichiro Sato

**Affiliations:** 1Graduate School of Chemical Science and Engineering, Hokkaido University, Sapporo 060-8628, Japan; erika_terada@eis.hokudai.ac.jp (E.T.);; 2Faculty of Engineering, Hokkaido University, Sapporo 060-8628, Japan; 3Research Center for Polymer Materials, School of Materials Science and Engineering, Changchun University of Science and Technology, Weixing Road 7989, Changchun 130012, China

**Keywords:** temperature-responsive polymers, poly(glycidyl ether), molecular dynamics simulation

## Abstract

Recently, experimental investigations of a class of temperature-responsive polymers tethered to oligooxyethylene side chains terminated with alkyl groups have been conducted. In this study, aqueous solutions of poly(glycidyl ether)s (PGE) with varying numbers of oxyethylene units, poly(methyl(oligooxyethylene)_n_ glycidyl ether) (poly(Me(EO)_n_GE)), and poly(ethyl(oligooxyethylene)_n_ glycidyl ether) (poly(Et(EO)_n_GE) (*n* = 0, 1, and 2) were investigated by all-atom molecular dynamics simulations, focusing on the thermal responses of their chain extensions, the recombination of intrapolymer and polymer–water hydrogen bonds, and water-solvation shells around the alkyl groups. No clear relationship was established between the phase-transition temperature and the polymer-chain extensions unlike the case for the coil–globule transition of poly(*N*-isopropylacrylamide). However, the temperature response of the first water-solvation shell around the alkyl group exhibited a notable correlation with the phase-transition temperature. In addition, the temperature at which the hydrophobic hydration shell strength around the terminal alkyl group equals the bulk water density ($T_{\mathrm{CRP}}$) was slightly lower than the cloud point temperature ($T_{\mathrm{CLP}}$) for the methyl-terminated poly(Me(EO)_n_GE) and slightly higher for the ethyl-terminated poly(Et(EO)_n_GE). It was concluded that the polymer-chain fluctuation affects the relationship between $T_{\mathrm{CRP}}$ and $T_{\mathrm{CLP}}$.

## 1. Introduction

Thermoresponsive polymers with a lower critical solution temperature (LCST) are attracting significant research interest. These polymers exhibit structural changes in response to external temperature change; thus, they are exploited for many applications, including drug delivery. Among the LCST-type thermoresponsive polymers, including poly(*N*-isopropylacrylamide) (PNIPAM) [1,2], poly(ethylene glycol) (PEG) [3], poly[poly(ethylene glycol) methyl ether methacrylate] [4,5,6], poly(glycidyl ether) (PGE) [7,8,9,10], polyisocyanates [11], poly(*N*,*N*-diethylacrylamide) [12], poly(*N*-vinylcaprolactam) [13,14], and poly [2-(dimethylamino)ethyl methacrylate] [15], PGE is known for its low cost, handling ease, and nontoxicity; thus, many PGE derivatives are commercially available. Additionally, PGE is widely employed in biomedicine for drug delivery [16,17] and as a scaffold material for tissue engineering [18,19]. Studies have shown that the LCST of PGE is controlled by its molecular weight [9] and molecular structure [8,10], as well as the balance between hydrophilic and hydrophobic side chains [10,20].

For example, Watanabe et al. reported that poly(glycidyl methyl ether), poly(ethyl glycidyl ether), and poly(ethoxyethyl glycidyl ether) exhibited LCST-type phase transitions from 14.6 °C to 57.7 °C when the hydrophobicity of their side chains was reduced [20]. In addition, we previously synthesized PGEs with various oligooxyethylene side chains terminated with an alkyl group and established that the LCSTs of poly(glycidyl methyl ether), poly(ethyl glycidyl ether), poly(2-methoxyethyl glycidyl ether), poly(2-ethoxyethyl glycidyl ether), and poly(2-(2-ethoxyethyl)ethyl glycidyl ether) phase transitions increased from 10.3 °C to 91.6 °C with a decrease in the hydrophobicity of their oligooxyethylene side chains [10]. However, despite the application prospects of PGE as a promising thermoresponsive material, the relationship between its temperature responsivity and polymer structure has not been adequately studied.

Molecular dynamics (MD) simulations have been employed to understand the response mechanisms of some thermoresponsive polymers, including PNIPAM [21,22,23], PGE [24], PEG [25], and poly(*N*-vinylcaprolactam) (PVCL) [26,27,28]. In addition, there is a research on the hydrophobic behavior of carbon nanoparticles using MD simulation in the study of their water solubility [29]. Conventionally, PNIPAM undergoes a coil–globule transition by the recombination of the polymer–water hydrogen bonds with the intrapolymer hydrogen bonds [21,22,23]. Conversely, the mechanism of the coil–globule transition in PGE has not yet been clarified. From a thermodynamic standpoint, hydrogen bonds between polymer chains and water molecules contribute favorable enthalpies to the free energy of mixing, whereas bonds between water molecules and polymer chains, particularly hydrophobic hydration shells formed around hydrophobic groups, promote the ordering of water molecules. Therefore, they contribute negatively to the mixing entropy. At relatively high temperatures, the entropy term, TS, becomes predominant, and the free energy of mixing becomes positive, which accounts for the insolubility of the polymer. The coil–globule structural transition in PNIPAM has been studied using MD simulations [21,22,23]. Additionally, Pica and Graziano reported that the solvent exclusion volume effect can be an indicator of the LCST phase transition [30,31]. PGE, which is actively studied in biomedicine, requires strict temperature control. If the LCST temperature can be predicted by MD simulations, the polymer structure can be designed according to the target temperature.

In this study, we performed MD simulations on six different types of side-chain structures investigated by Isono and Satoh et al. [10] and examined the effect of the number of oxyethylene units on the LCST. We found that the hydrophobic shell intensity of the sidechain terminal alkyl groups well-correlated with LCST.

## 2. Computational Method

### 2.1. Initial Structure of the Polymers

Models of poly(glycidyl methyl ether) (poly(MeGE)), poly(ethyl glycidyl ether) (poly(EtGE)), poly(2-methoxyethyl glycidyl ether) (poly(MeEOGE)), poly(2-ethoxyethyl glycidyl ether) (poly(EtEOGE)), poly(2-(2-methoxyethoxy)ethyl glycidyl ether) (poly(MeEO_2_GE)), and poly(2-(2-ethoxyethyl)ethyl glycidyl ether) (poly(EtEO_2_GE)) were prepared by the random addition of monomers with *d* and *l* structures, mimicking the tacticity of the whole chain (Figure 1). These polymer structures were optimized by the molecular mechanics 2 (MM2) method included in the chem3D 19.0 package of PerkinElmer (Waltham, MA, USA). All the polymer molecular weights (MW) were set at ca. 2500, and 5000.

### 2.2. Atomic Charges and Force Fields

The atomic partial charges for the six polymer species were obtained through the AM1-BCC protocol [32,33] implemented in the antechamber program [34] of AMBER 14. All other force-field parameters for the polymers were acquired using the general AMBER force field (GAFF) version 1.7 [34]. The GAFF was chosen because it has been demonstrated to be accurate for the simulation of most organic molecules. The detailed force field parameters are provided in gaff.dat [34].
(1)$$\begin{array}{c} V_{\mathrm{AMBER}}=\sum_{i}^{n_{\mathrm{bonds}}}b_{i}{\left(r_{i}-r_{i,\mathrm{eq}}\right)}^{2}+\sum_{i}^{n_{\mathrm{angles}}}a_{i}{\left(\theta_{i}-\theta_{i,\mathrm{eq}}\right)}^{2}+\sum_{i}^{n_{\mathrm{dihedrals}}}\sum_{n}^{n_{i,\max}}(V_{i,\;n}/2)\left[1+\cos\left(n\emptyset_{i}-\gamma_{i,\;n}\right)\right]\;\\ +\sum_{i<j}^{n_{\mathrm{atoms}}}\prime \left(\frac{A_{ij}}{r_{ij}^{12}}-\frac{B_{ij}}{r_{ij}^{6}}\right)+\sum_{i<j}^{n_{\mathrm{atoms}}}\prime \frac{q_{i}q_{j}}{4\pi \varepsilon_{0}r_{ij}}, \end{array}$$

### 2.3. Solvent Water Addition into a Simulation Box

Each polymer was centered in a cubic simulation box under periodic boundary conditions. Over 2000 TIP3P-modeled water molecules were added with a margin of 10 Å along each dimension in Appendix A.

### 2.4. Energy Minimization and Temperature Equilibration

The whole system consisted of a polymer, and the water molecules were energy-minimized and equilibrated with the sander program implemented in the AMBER system before the MD simulations. The energy minimization was performed in two steps. First, the solvent molecules were energy-minimized using the steepest decent method for 1000 steps and the conjugate gradient method for 1500 steps while keeping the solute molecular chains restrained with a force of 500 kcal mol^−1^ Å^−2^. Second, the restraint was eliminated, and the entire system, including the polymer chain, was energy-minimized using the steepest decent method for 1000 steps and the conjugate gradient method for 1500 steps as in the first step.

Subsequently, the energy equilibration was performed in two steps. First, the system was gradually heated to an assigned temperature, 278, 300, 323, 343, and 368 K, under constant volume boundary conditions for 10,000 steps, with a time step of 2 fs while maintaining the 10 kcal mol^−1^ Å^−2^ constraint on the solute polymer chains. In the second step, the restraint was removed, and the entire system was equilibrated under periodic boundary conditions at a constant pressure of 1 atm for 50,000 steps, with a time step of 2 fs while maintaining the specified temperature.

The Langevin method was employed for temperature control, and isotropic scaling was employed for pressure control. The SHAKE algorithm was applied to maintain constant hydrogen atom bond lengths. A cutoff length of 9.0 Å and particle mesh Ewald (PME) sums were applied for the electrostatic interactions.

### 2.5. MD Simulation

After confirming the equilibrium of the whole system, MD simulations were performed for 20 ns or 100 ns, with a time step of 2 fs using a pmemd program of AMBER 14. All the MD simulations were conducted in NPT (number of molecules (N), pressure (P), and temperature (T) are conserved) ensembles to capture the dynamics and structural evolution of both the polymer and water molecules at five different temperatures (278, 300, 323, 343, and 368 K). The simulations were carried out at atmospheric pressure (1 atm). The trajectories were outputted at 2.0 ps intervals.

## 3. Analysis Method

### 3.1. Radius of Gyration

The structural evolution of each polymer was studied along the time course of the stored atomic trajectories by analyzing the radius of gyration (*R*_g_). *R*_g_, which is a measure of the extension degree of a polymer chain, was calculated using the equation below:(2)$$R_{g}^{2}=\frac{1}{N^{\prime }}\sum_{n}\langle {\left(R_{n}-R_{c}\right)}^{2}\rangle ,$$
where *R*_c_ is the center of mass of the polymer chain, and *N* is the number of atoms included in the polymer chain. The sum for all the atoms in the chain was obtained.

### 3.2. Radial Distribution Function

The radial distribution function (RDF), $g\left(r\right)$, represents the number density of the particles existing in the spherical shells between distances *r* and r + dr from a given reference particle. It can be expressed as:(3)$$g\left(r\right)=\frac{\left\langle n(r)\right\rangle }{4\pi r^{2}\mathrm{dr}\rho }$$
where $n\left(r\right)$ indicates the number of particles of the spherical shell at distances of r and r+dr from the given reference particle. ρ is the average number density of the particles in the bulk condition. The $g\left(r\right)$ function is zero when the interparticle distance, r, approaches zero because repulsive force prevents the atoms from closely approaching each other. Additionally, when distance r is infinite, the local density in the shell is equivalent to the mean bulk density averaged over the whole volume; therefore, $g\left(r\right)$ is standardized to 1.

The RDF was calculated to determine the difference in the structural arrangements of the polymer atoms and water molecules at 278, 300, 323, 343, and 368 K. The RDFs around the carbon atom of the side chain and for the water–oxygen atom were carefully investigated for six different kinds of polymers.

## 4. Results and Discussion

### 4.1. Temperature Dependence of R_g_

In PNIPAM, the LCST-type phase transition and coil–globule structural transitions are known to be correlated. The occurrence of the coil–globule structural transition can be determined using the *R*_g_, which represents the degree of chain extension. Therefore, the temperature dependence of *R*_g_ was analyzed to observe the structural transition behaviors of poly(MeGE), poly(EtGE), poly(MeEOGE), poly(EtEOGE), poly(MeEO_2_GE), and poly(EtEO_2_GE). Five polymers other than poly(MeEO_2_GE) have been reported to exhibit phase transitions [10]. In this study, MD simulations were performed with molecular weights of 2500 and 5000 for all polymers. The degrees of polymerization with molecular weights of 2500 for poly(MeGE), poly(EtGE), poly(MeEOGE), poly(EtEOGE), poly(MeEO_2_GE), and poly(EtEO_2_GE) were 28, 24, 19, 17, 14, and 15, respectively. Additionally, the degrees of polymerization with molecular weights of 5000 for poly(MeGE), poly(EtGE), poly(MeEOGE), poly(EtEOGE), poly(MeEO_2_GE), and poly(EtEO_2_GE) were 55, 47, 37, 33, 27, and 29, respectively. The averaged trajectories obtained from two MD simulations were employed for the analysis. Figure 2 and Appendix A show the *R*_g_ distributions for six different kinds of polymers having molecular weights 2500 obtained with their time courses for 20 ns. The *R*_g_ distributions of poly(EtGE), poly(EtEOGE), and poly(EtEO_2_GE) with ethyl terminal groups were relatively unimodal across the temperature range, whereas those of poly(MeGE) and poly(MeEOGE) with methyl terminal groups were multimodal or broadened in the low-to-medium temperature range. Table 1 summarizes the peak values of the *R*_g_ distribution at each temperature and the corresponding experimental cloud point (CLP) values. The experimental CLP temperatures ($T_{\mathrm{CLP}}$) are referenced from the results obtained using the molecular weight of ca. 5000 [10]. Figure 3 and Figure 4 shows the *R*_g_ peak values versus the temperatures in Table 1. No significant change was observed in the main peak values of *R*_g_ above and below $T_{\mathrm{CLP}}$ for any of the polymers. This suggests that the coil–globule structural transition was not a prerequisite for the LCST-type phase transition in the PGEs with oxyethylene units. For the standard deviation (SD), the PGE with side-chain ethyl groups tended to have a smaller SD value than those with side-chain methyl group. This implies that the PGE with ethyl groups at the side-chain ends are bulkier than those with methyl groups, and thus, the movement of the main chain structure should be suppressed. Notably, the difference in the bulkiness of the terminal structure affected the distribution of *R*_g_. Appendix A shows snapshots of each polymer corresponding to the main peak values of *R*_g_ at temperatures below and above $T_{\mathrm{CLP}}$. No clear coil–globule structural transitions were observed above and below $T_{\mathrm{CLP}}$ for any of the polymers, as evidenced by the analysis of *R*_g_. It was difficult to observe structural transitions from the *R*_g_ distributions and snapshots for six different kinds of polymers having molecular weights of 5000 obtained with their time courses for 100 ns in Figure 5 and Appendix A. The *R*_g_ distributions of polymers with the molecular weight 5000 tended to have larger width than those with molecular weight 2500, which should be attributed to the greater fluctuations caused by the doubling of the molecular weight.

### 4.2. Degree of Chain Extensions and Their Fluctuations

To investigate whether the polymer is in the coil or globule state, the ratio of the maximum length of the main chain $\left(L_{\max}\right)$ to the time average length $\left(\bar{L}\right)\;$obtained from the MD simulation performed with a molecular weight of 2500 was calculated. The time dependence and distribution of $\bar{L}$ for the six polymers are shown in Appendix A. For freely jointed chains or ideal chains with the degree of polymerization, *N*, the following equation holds:(4)$$\frac{\bar{L}}{L_{\max}}=\frac{1}{\sqrt{N}}.$$

If the ratio is larger than $1/\sqrt{N}$, the polymer can be roughly considered to be in the coil state, and if it is smaller, it can be considered to be in the globule state [35]. Table 2 summarizes the values of $\bar{L}/L_{\max}$ and $1/\sqrt{N}$ for all the polymers. For poly(MeGE) (28-mers) and poly(EtGE) (24-mers), which had relatively large degrees of polymerization, the values of $\bar{L}/L_{\max}\;$were smaller than the corresponding $1/\sqrt{N}$ values in almost all temperature ranges, suggesting that they consistently adopted globule structures. Poly(MeEOGE) (19-mers) exhibited larger values of $\bar{L}/L_{\max}\;$than $1/\sqrt{N}$ in all temperature ranges; thus, it could be considered to consistently be in the coil state. Conversely, poly(EtEOGE) (17-mers) exhibited smaller values of $\bar{L}/L_{\max}$ than $1/\sqrt{N}$ at 300, 323, and 343 K; thus, it was expected to be in the globule state. Poly(MeEO_2_GE) (14-mers) and poly(EtEO_2_GE) (15-mers) with relatively small degrees of polymerization exhibited larger values of $\bar{L}/L_{\max}$ than $1/\sqrt{N}$; thus, they were expected to be in the coil state in all temperature ranges. The coefficient of variation (CV) was calculated to compare the fluctuation of $\bar{L}/L_{\max}$ for the six polymers without the effect of different degrees of polymerization. The CV is the SD normalized by the mean value. It is a dimensionless number used to evaluate the relative relationship of the SDs among the data with different mean values. When the CV values between poly(MeGE) and poly(EtGE), poly(MeEOGE) and poly(EtEOGE), and poly(MeEO_2_GE) and poly(EtEO_2_GE) were compared, the polymer with the methyl group at the side-chain ends tended to exhibit higher values than those for the polymer with the ethyl group at the side-chain ends. According to the *R*_g_ measurement results, the magnitude of fluctuation was different between the methyl and ethyl side-chain ends; however, it was unclear whether the difference was due to the main chain or the side chain. This suggests that the polymer main chain structure with the methyl group fluctuated more than that with the ethyl group. Table 3 shows the ratio of the maximum length of the main chain $\left(L_{\max}\right)$ to the temperature average length $\left({\left\langle \bar{L}\right\rangle }_{\mathrm{temp}}\right)$, SD, and CV for the six polymers at five temperatures. Resultantly, except for poly(MeEO_2_GE), the CV values of the polymers with terminal methyl groups were larger than those of the polymers with terminal ethyl groups. This indicates that the main-chain structural fluctuation of the polymer with a terminal methyl group on the side chain should be larger than that of the polymer with a terminal ethyl group on the side chain.

It has been reported that the $T_{\mathrm{CLP}}$ of poly(MeEOGE) decreases as the molecular weight increases [10]. The $\bar{L}/L_{\max}$ values of poly(MeEOGE) for molecular weights of 1250 and 5000 were calculated, and the polymer-chain extension was investigated. Table 4 summarizes the results of the $\bar{L}/L_{\max}$ and $1/\sqrt{N}$ calculations for poly(MeEOGE) with molecular weights of 1250, 2500, and 5000. As the molecular weight increased, the value of $\bar{L}/L_{\max}$ tended to decrease. For the molecular weight of 1250, the values of $\bar{L}/L_{\max}$ were larger than the corresponding $1/\sqrt{N}$ values in all temperature regions except for 278 K. This indicated that poly(MeEOGE) with a molecular weight of 1250 exhibited a coil structure. For the molecular weight of 5000, the values of $\bar{L}/L_{\max}$ were smaller than the $1/\sqrt{N}$ values in all temperature regions, indicating that the poly(MeEOGE) with a molecular weight of 5000 had a globule structure. A similar tendency was observed for poly(EtEOGE) (Appendix A). These results suggested that the smaller the degree of polymerization of PGE, the lower the probability that it exhibits the coil structure, and the larger the degree of polymerization, the higher the probability that it exhibits the globule structure. The larger the degree of polymerization of poly(MeEOGE), the higher its tendency to adopt the globule structure. This implies that the change from the coil to the globule state with increasing molecular weight was related to the change in the $T_{\mathrm{CLP}}$. This relation will be explained by RDF analysis.

### 4.3. The Side-Chain Lengths and Their Fluctuations

The results of the $\bar{L}/L_{\max}$ calculations suggested that the main chain structure of PGE with methyl terminal groups fluctuated more than that of the PGE with ethyl terminal groups at the side-chain ends. To further investigate if the fluctuation of the side-chain structure affects the polymer fluctuation, the side-chain lengths (i.e., the lengths between the main-chain carbon atom and the side-chain terminal carbon atom) of six polymers with molecular weights of 2500 were obtained using the time-averaged values of the side-chain length in the MD simulations. Table 5 summarizes the time-averaged side-chain lengths, and their temperature-averaged values, SDs, and CVs, as well as the calculated side-chain lengths at each temperature, are shown in Appendix A. When comparing the CV values, when the side chain was short, as in poly(MeGE) and poly(EtGE), no difference in the magnitude of the fluctuations was observed. However, when the side chain was long, as in poly(MeEOGE), poly(EtEOGE), poly(MeEO_2_GE), and poly(EtEO_2_GE), the PGE with the methyl group exhibited a higher magnitude of fluctuation in the side-chain structure than those with ethyl group. This suggested that the fluctuation of the side-chain structure affected the fluctuation of the polymer structure.

### 4.4. RDF of Hydrophobic Solvation

Since the structural transition corresponding to the LCST-type phase transition was not observed by *R*_g_ analysis, an alternative physical quantity that reflects CLP was investigated. The RDF is utilized to explore the local order of water molecules surrounding polymers. Therefore, the RDF for the oxygen atoms of waters with polymers having carbon atoms at the side-chain ends was investigated. All the results are based on the average values obtained from two MD simulations conducted with a molecular weight 2500. Figure 6 shows the water–oxygen RDF measured using the carbon atoms at all the side-chain ends of poly(MeGE), poly(EtGE), poly(MeEOGE), poly(EtEOGE), poly(MeEO_2_GE), and poly(EtEO_2_GE) at 278, 300, 323, 343, and 368 K, respectively. The first peaks in the water–oxygen RDF spectra were observed at around 3.6 Å for all polymers; thus, the first hydrophobic hydration shells are formed at 3.6 Å from carbon atoms at the side-chain ends. Thereafter, the temperature dependence of the peak intensity in the first hydrophobic hydration shell was investigated. As shown in Figure 7, the peak intensity of the hydrophobic hydration shell decreased during the heating process. Furthermore, the crossing point temperature ($T_{\mathrm{CRP}}$) at which the peak intensity decreased below $g\;\left(r\right)=1.0$ was analyzed for the six types of polymers. The $T_{\mathrm{CRP}}$ values are summarized together with the $T_{\mathrm{CLP}}$ values reported in the experiments [10] in Table 6. The $T_{\mathrm{CRP}}$ values for poly(MeGE), poly(EtGE), poly(MeEOGE), poly(EtEOGE), and poly(EtEO_2_GE) appeared at 326, 291, 349, 334, and 348 K, respectively, and they were close to the experimental $T_{\mathrm{CLP}}$ values. These results indicated that the hydrophobic hydration shell strength at the end of the polymer side chain correlated with $T_{\mathrm{CLP}}$. By examining the side-chain end structures, we established that the $T_{\mathrm{CRP}\;}$ obtained by simulation for poly(MeGE) and poly(MeEOGE) with methyl-side-chain ends were lower than the experimentally obtained $T_{\mathrm{CLP}}$ values. Conversely, the $T_{\mathrm{CRP}}$ obtained by simulation for poly(EtGE), poly(EtEOGE), and poly(EtEO_2_GE) with ethyl-side-chain ends were higher than the experimentally obtained $T_{\mathrm{CLP}}$ values. In this study, MD simulations were performed with a single polymer chain, and the $T_{\mathrm{CRP}}$ should be higher than the experimental $T_{\mathrm{CLP}}$ because the polymer concentration in the MD simulation is lower than that in the actual experiment. In the experiments, $T_{\mathrm{CLP}}$ tends to increase as the concentration of the polymer solution decreases. This is because the collision frequency between the polymers is reduced, and aggregation is less likely to occur. Contrarily, for the PGE with methyl groups at the side-chain ends, $T_{\mathrm{CRP}}$ was lower than $T_{\mathrm{CLP}}$. This relationship is plotted in Figure 8. The PGE with methyl terminal groups exhibited large fluctuations in the main-chain and side-chain structures, as evidenced by the results of *R*_g_, $\bar{L}/L_{\max}$, and time-averaged side-chain length measurements. Therefore, it is expected that water molecules enter the gaps between the adjacent side chains of the PGE with hydrophobic methyl groups, while the proportion of the side-chain end occupying the polymer surface decreases, and the proportion of the hydrophilic ether part occupying the polymer surface increases (Figure 9). $T_{\mathrm{CRP}}$ only reflects the aqueous environment at the end of the side chain and not the entire polymer surface. This suggests that the $T_{\mathrm{CRP}}$ of the PGE with methyl groups at the side-chain ends was lower than the $T_{\mathrm{CLP}}$.

In addition, to investigate if the molecular weight affects the RDF, the $T_{\mathrm{CRP}}$ values for poly(MeEOGE) and poly(EtEOGE) with molecular weights of 1500, 2500, and 5000 were analyzed. Figure 10 shows that as the molecular weight increased, $T_{\mathrm{CRP}}$ decreased. It has been reported that the phase-transition temperature of poly(MeEOGE) tends to decrease with increasing molecular weight [10], and a similar result was observed in our $T_{\mathrm{CRP}}$ analysis in the MD calculations. This suggests that as the molecular weight increased, the hydrophobicity of the polymer surface increased because of the hydrophobic side chain, resulting in a decrease in $T_{\mathrm{CLP}}$.

### 4.5. Numbers of Hydrogen Bonds between the Polymer and Water Molecules

In the previous section, the hydrophobic hydration shells were analyzed, after which the hydrogen bonds between the polymer oxygen atoms and water molecules were investigated to determine the contribution of the hydrophilic ether oxygen to the LCTS-type phase transition. The average number of hydrogen bonds between the oxygen atoms in the main chains or the side chains and the hydrogen atoms in water are listed in Table 7. Figure 11 and Figure 12 show the average numbers of hydrogen bonds between the polymer and water molecules in the main-chain oxygen atoms or the side-chain oxygen atoms per oxygen in the polymer residue, ($\left\langle N_{\mathrm{main}\;\mathrm{chain}\;\mathrm{HB}}\right\rangle$), or ($\left\langle N_{\mathrm{side}\;\mathrm{chain}\;\mathrm{HB}}\right\rangle$), versus temperature. The average numbers of hydrogen bonds between the polymer and water molecules in the main-chain oxygen atoms tend to decrease as the temperature increases for all polymers, and similar results were obtained for the side-chain oxygen atoms. In addition, the average number of hydrogen bonds between the water and oxygen atoms in the side chain tended to be higher than that of the bonds between the water and oxygen atoms in the main chain.

Furthermore, a relatively large decrease in the number of hydrogen bonds around $T_{\mathrm{CLP}}$ or $T_{\mathrm{CRP}}$ was observed. Although not shown in Table 7, the average number of hydrogen bonds between the intrapolymer was almost zero at any temperature for all polymers. The average numbers of hydrogen bonds within an intrapolymer are shown in Appendix A. This suggests that the recombination of intrapolymer hydrogen bonds and polymer–water hydrogen bonds for the OEO ether group of the PGE side chain had not occurred. N-substituted acrylamide-based polymers, such as PNIPAM, have amide groups with a proton donor and acceptor, and they promote the recombination of hydrogen bonds between an intrapolymer and between polymer and water molecules [36,37]. However, since PGE has ether groups with only a proton acceptor, the hydrogen bonds between the intrapolymer should be weak; therefore, no hydrogen bond recombination was observed. In addition, it has been reported that hydrogen bonding does not affect the LCST-type transition in poly(oligo(ethylene glycol)methyl ether methacrylate) with an ether group [38]. This suggests that the weak hydrogen bonding in the PGE with ether groups led to the absence of a clear coil–globule structural transition.

## 5. Conclusions

In this study, we performed MD simulations of PGE with six different oligooxyethylene side chains terminated with alkyl groups in aqueous solutions to investigate the effects of side-chain structures on LCST-type phase transition. To investigate if the coil–globule structural transition occurs, we analyzed the *R*_g_, which indicates the degree of polymer chain extension, and the $\bar{L}/L_{\max}$, which indicates the degree of polymer shrinkage. Comparing the SD of *R*_g_ and the CV values of $\bar{L}/L_{\max}$ and the side-chain lengths for the six polymers, the PGE with a methyl group at the end of the side chain revealed larger values than the PGE with an ethyl group, suggesting that the PGE with terminal methyl groups on the side chain has larger fluctuations both in the main and side chains than the PGE with terminal ethyl groups on the side-chain. Although we did not observe clear coil–globule structural transition in any PGEs, we found that, as the degree of polymerization increased, the globule structure became easier to obtain. To investigate if the coil–globule transition occurs in more detail, we will analyze the eigenvalues of the radius of the gyration tensor in our future study.

Next, we focused on the RDF as another physical quantity that indicates the distribution of water molecules around the polymer and calculated the RDF for the oxygen atoms of waters measured from carbon atoms at the side-chain ends of the polymers. The result showed that hydrophobic hydration shells were formed around 3.6 Å from the carbon atoms at the end of the polymer side chains in all six polymers. We investigated the temperature dependence of the peak intensity of the hydrophobic hydration shells and found that the crossing point temperatures ($T_{\mathrm{CRP}}$) at which the peak intensity was below $g\;\left(r\right)=1.0$ were close to the CLP temperatures ($T_{\mathrm{CLP}}$) obtained in the experiments [10] for all polymers. The phase-transition temperature of the PGE was correlated with the intensity of the primary hydrophobic hydration shell of the side-chain terminal alkyl carbon. Furthermore, the $T_{\mathrm{CRP}}$ of the PGE with terminal methyl groups on the side chain was lower than the experimental $T_{\mathrm{CLP}}$, while the $T_{\mathrm{CRP}}$ for the PGE with terminal ethyl groups on the side chain was higher than the experimental $T_{\mathrm{CLP}}$. In this work, we performed MD simulations with a single polymer chain; therefore, the $T_{\mathrm{CRP}}$ should be higher than the experimental $T_{\mathrm{CLP}}$ because the polymer concentration is lower than that in the actual experiment. We consider that the magnitude of the polymer fluctuation affects the relationship between the $T_{\mathrm{CRP}}$ and the $T_{\mathrm{CLP}}$. In other words, since the PGE with terminal methyl groups on the side chain tends to have larger polymer fluctuations than the PGE with terminal ethyl groups on the side chain, as evidenced by the results of *R*_g_, $\bar{L}/L_{\max}$, as well as time-averaged side-chain length measurements, we assume that water molecules entered the gaps between the adjacent side chains of PGE with hydrophobic methyl groups. Consequently, the proportion of the hydrophilic ether part occupying the polymer surface increased. Thus, the $T_{\mathrm{CRP}}$ of PGE with methyl groups at the side-chain ends was lower than the $T_{\mathrm{CLP}}$ value. In the future, we will expand our research to encompass multiple polymer chains to investigate interpolymer interactions.

## Figures and Tables

**Figure 1 nanomaterials-13-01628-f001:**
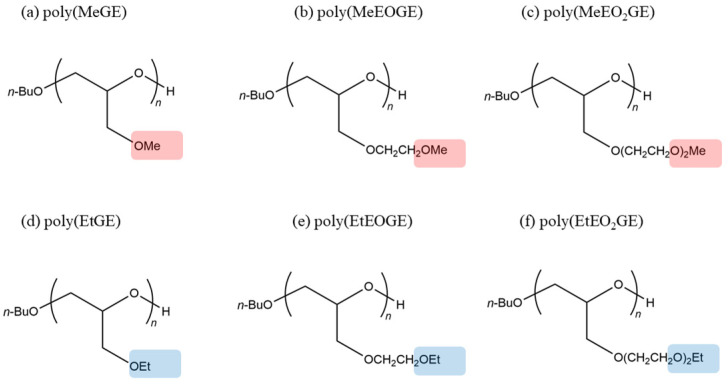
Chemical structures of (**a**) poly(MeGE), (**b**) poly(MeEOGE), (**c**) poly(MeEO_2_GE), (**d**) poly(EtGE), (**e**) poly(EtEOGE), and (**f**) poly(EtEO_2_GE). Red marker shows methyl group and blue marker shows ethyl group.

**Figure 2 nanomaterials-13-01628-f002:**
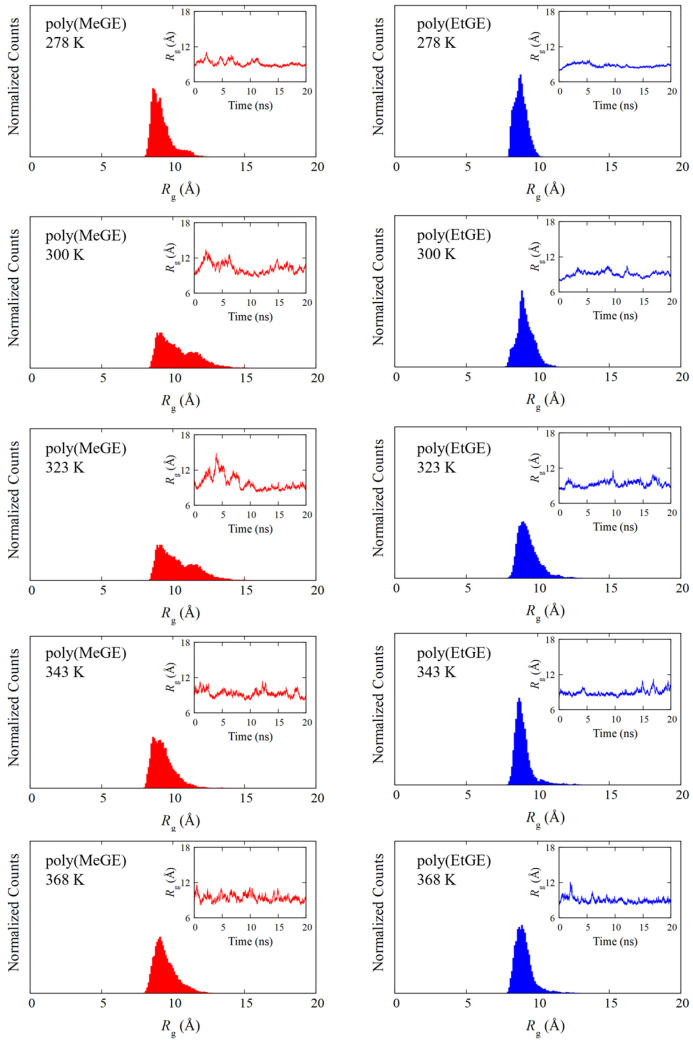
*R*_g_ distribution for poly(MeGE)_2.5k_ (**left**) and poly(EtGE)_2.5k_ (**right**). The inset is the corresponding time dependence of *R*_g_.

**Figure 3 nanomaterials-13-01628-f003:**
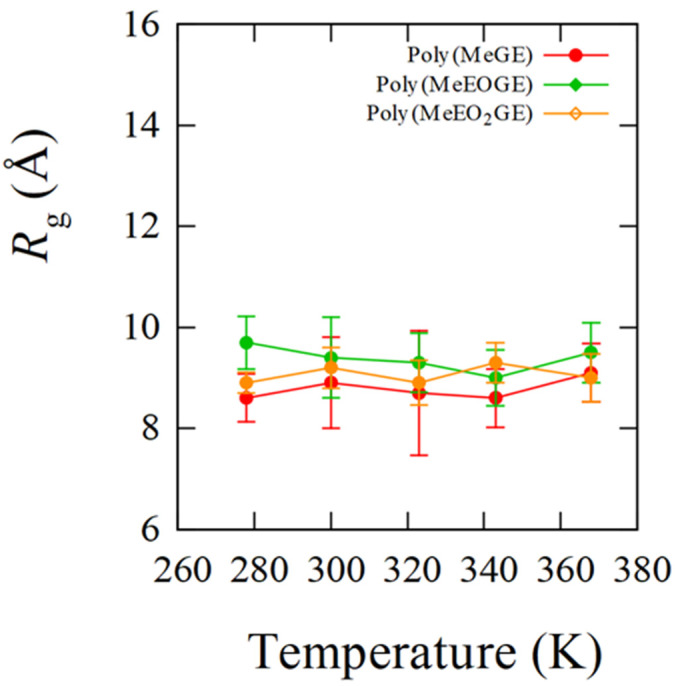
*R*_g_ peaks plotted versus temperatures for poly(MeGE)_2.5k_, poly(MeEOGE)_2.5k_, and poly(MeEO_2_GE)_2.5k_ with error bars.

**Figure 4 nanomaterials-13-01628-f004:**
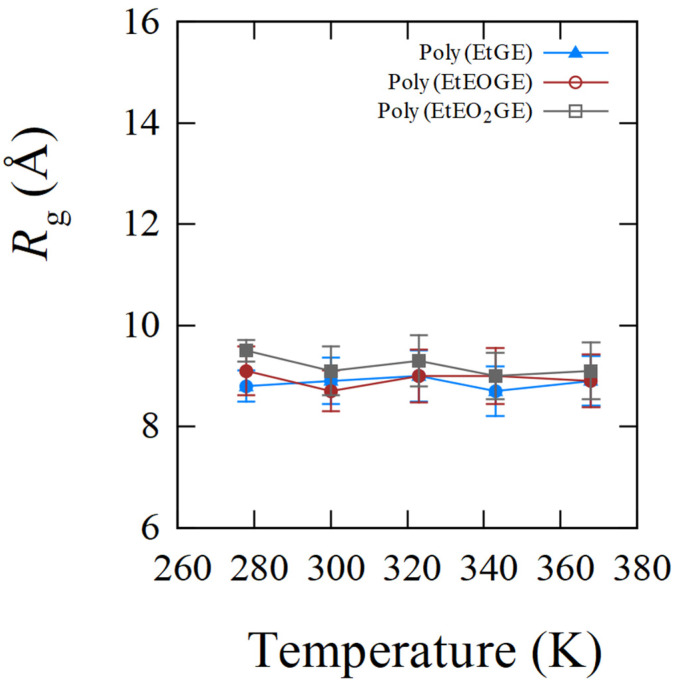
*R*_g_ peaks plotted versus temperatures for poly(EtGE)_2.5k_, poly(EtEOGE)_2.5k_, and poly(EtEO_2_GE)_2.5k_ with error bars.

**Figure 5 nanomaterials-13-01628-f005:**
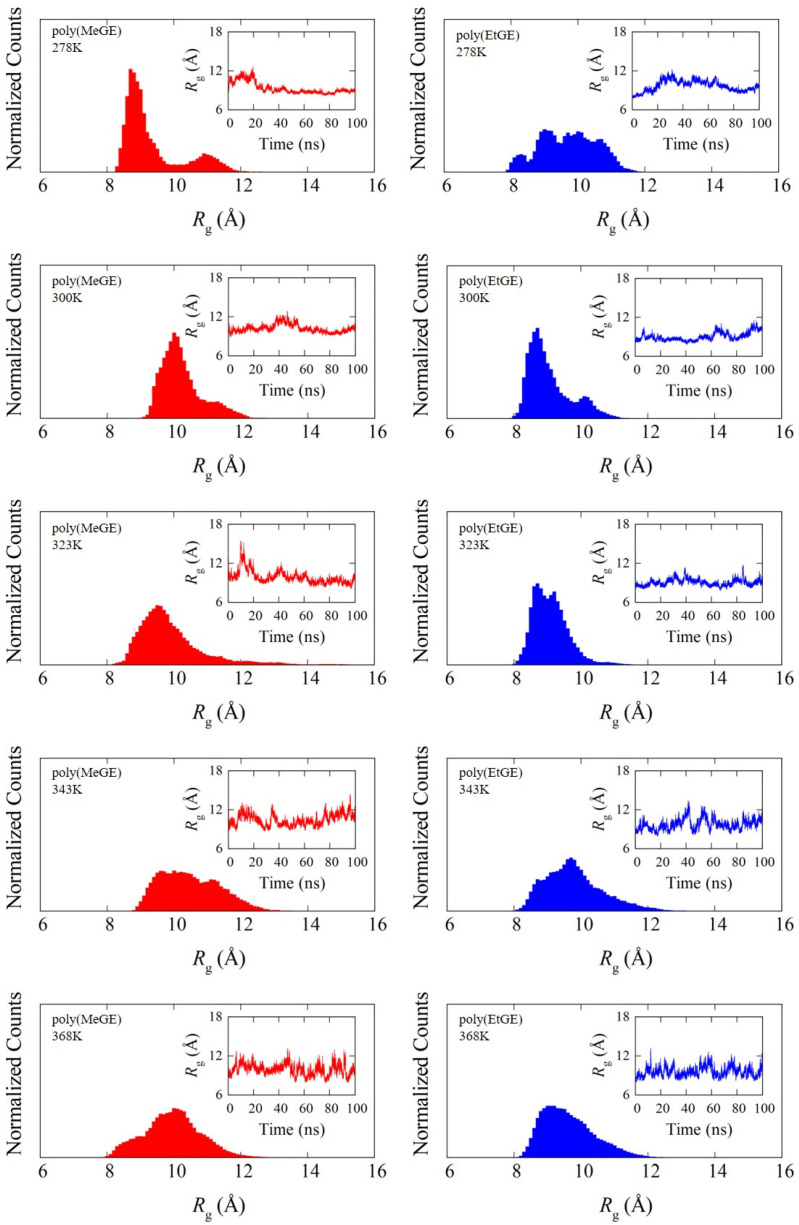
*R*_g_ distribution for poly(MeGE)_5k_ (**left**) and poly(EtGE)_5k_ (**right**). The inset is the corresponding time dependence of *R*_g_.

**Figure 6 nanomaterials-13-01628-f006:**
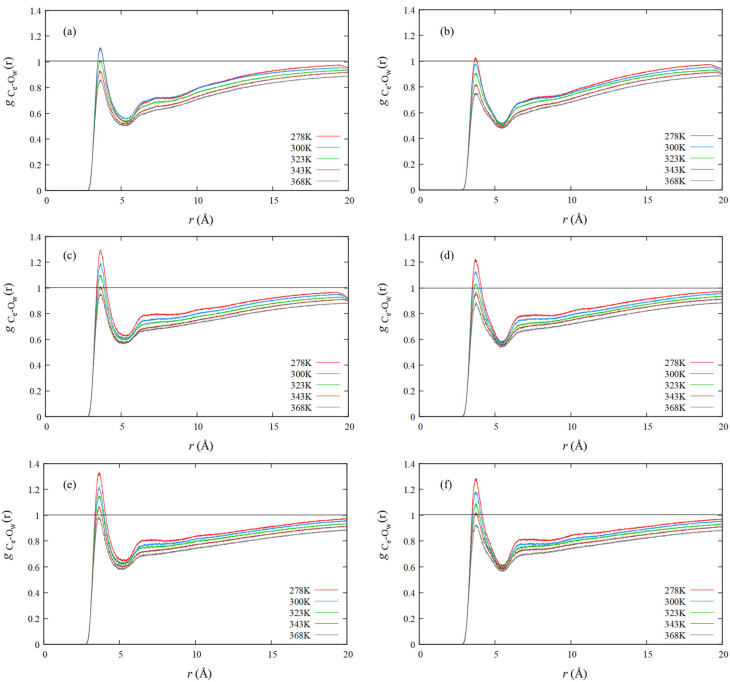
RDFs of (**a**) poly(MeGE)_2.5k_, (**b**) poly(EtGE)_2.5k_, (**c**) poly(MeEOGE)_2.5k_, (**d**) poly(EtEOGE)_2.5k_, (**e**) poly(MeEO_2_GE)_2.5k_, and (**f**) poly(EtEO_2_GE)_2.5k_ for O atoms of water measured from C atoms at the side-chain ends.

**Figure 7 nanomaterials-13-01628-f007:**
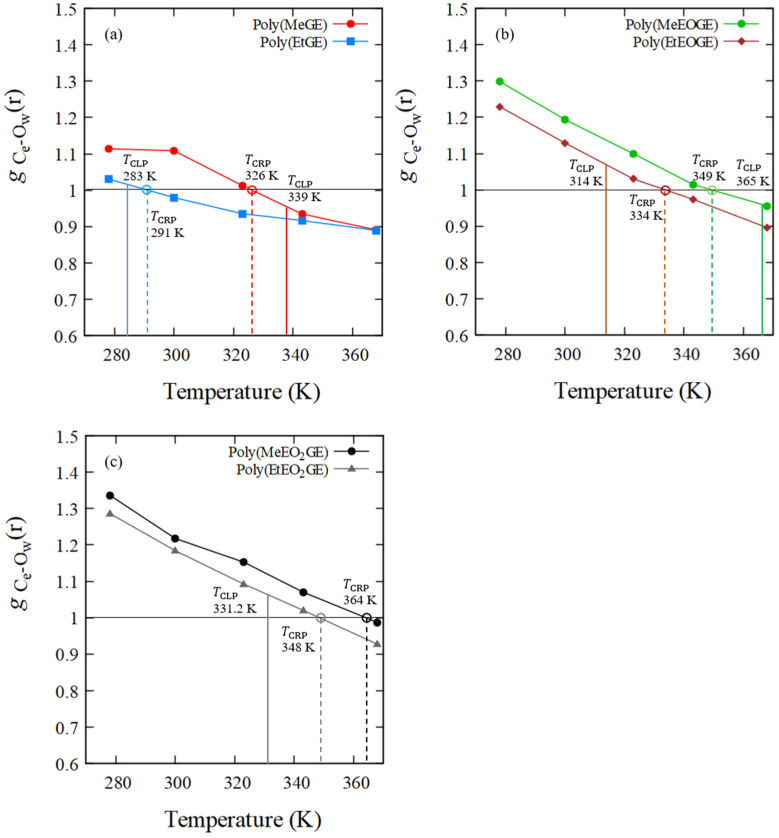
Temperature response of the first peak intensity in the RDFs of O atoms (water) observed from C atoms at the side-chain ends, (**a**) poly(MeGE)_2.5k_ and poly(EtGE)_2.5k_, (**b**) poly(MeEOGE)_2.5k_ and poly(EtEOGE)_2.5k_, and (**c**) poly(MeEO_2_GE)_2.5k_, and poly(EtEO_2_GE)_2.5k_. The solid line shows $T_{\mathrm{CLP}}$ and the broken line shows $T_{\mathrm{CRP}}$.

**Figure 8 nanomaterials-13-01628-f008:**
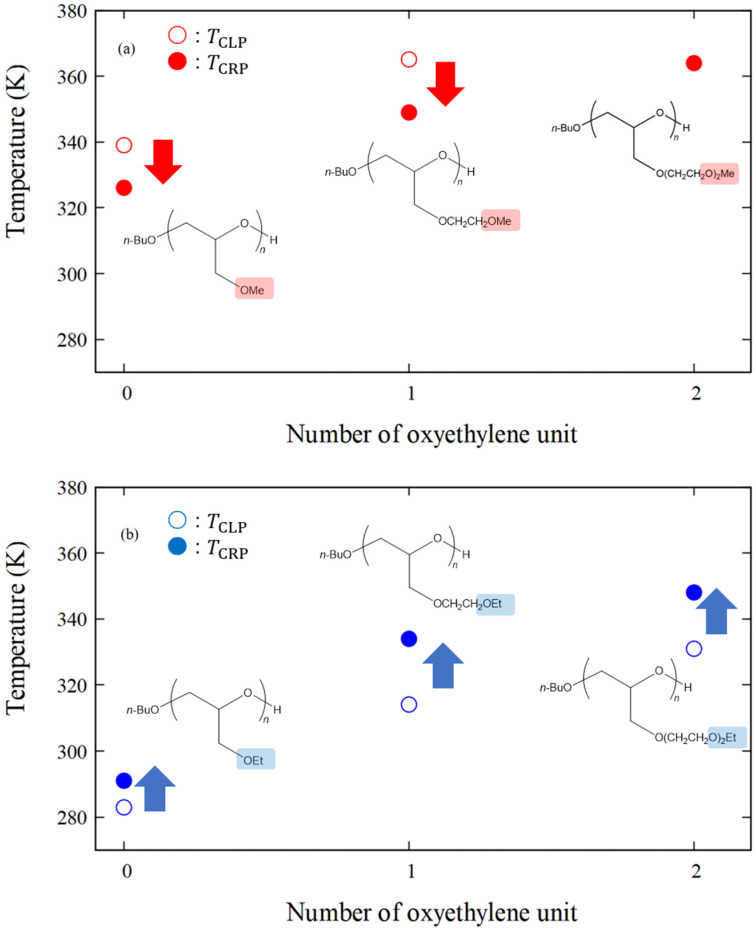
Plots of $T_{\mathrm{CRP}}$ and $T_{\mathrm{CLP}}$ vs. number of oxyethylene units for methyl-terminated poly(Me(EO)_n_GE) (**a**) and ethyl-terminated poly(Et(EO)_n_GE) (**b**). Open and filled circles correspond to $T_{\mathrm{CLP}}$ and $T_{\mathrm{CRP}}$, respectively. Red arrows mean the $T_{\mathrm{CRP}}$ is lower than the $T_{\mathrm{CLP}}$, and blue arrows mean the $T_{\mathrm{CRP}}$ is higher than the $T_{\mathrm{CLP}}$.

**Figure 9 nanomaterials-13-01628-f009:**
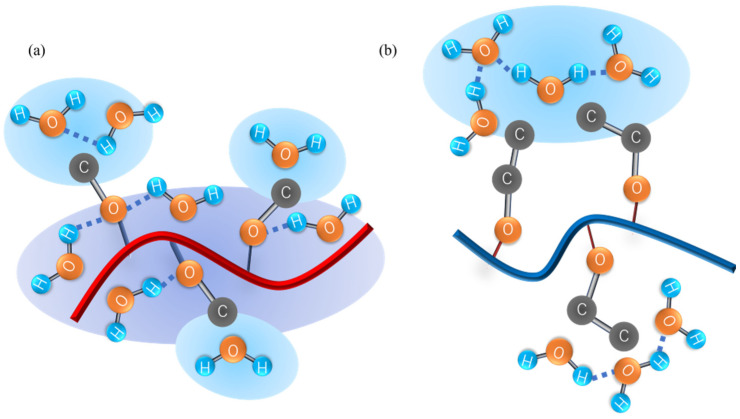
Schematic illustration of (**a**) poly(Me(EO)_n_GE) and (**b**) poly(Et(EO)_n_GE) in water.

**Figure 10 nanomaterials-13-01628-f010:**
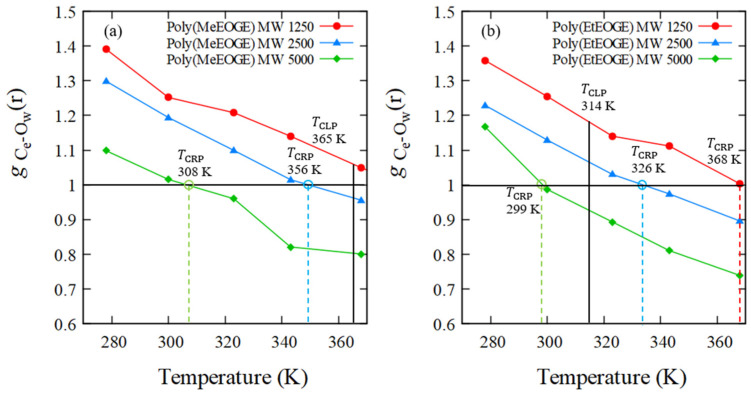
Temperature response of the first peak intensity in the RDFs of O atoms (water) observed from C atoms at the side-chain ends in (**a**) poly(MeEOGE) and (**b**) poly(EtEOGE) for MW 1250, 2500, and 5000. The solid line shows $T_{\mathrm{CLP}}$ and the broken line shows $T_{\mathrm{CRP}}$.

**Figure 11 nanomaterials-13-01628-f011:**
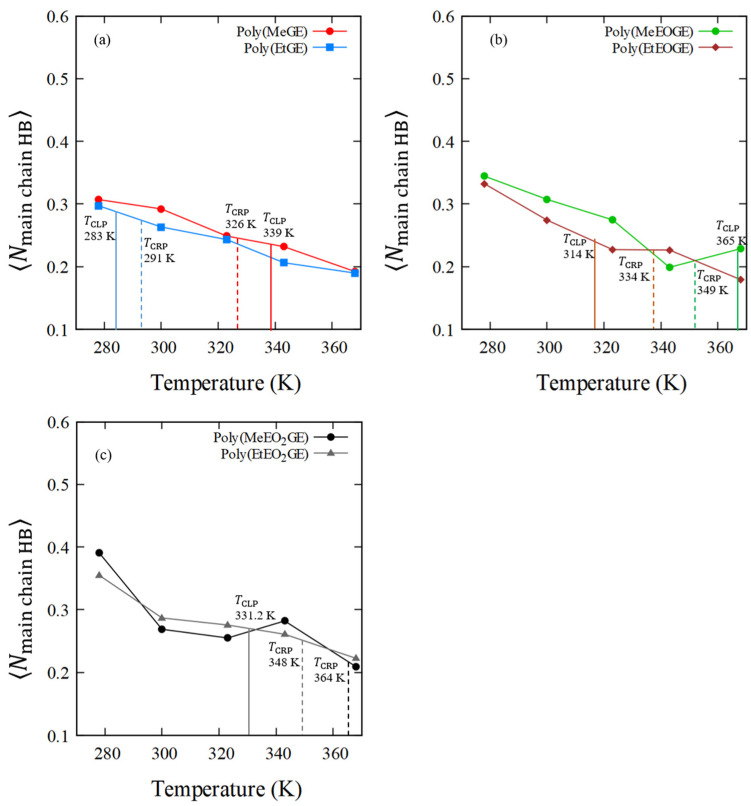
$\left\langle N_{\mathrm{main}\;\mathrm{chain}\;\mathrm{HB}}\right\rangle$ of (**a**) poly(MeGE)_2.5k_ and poly(EtGE)_2.5k_, (**b**) poly(MeEOGE)_2.5k_ and poly(EtEOGE)_2.5k_, and (**c**) poly(MeEO_2_GE)_2.5k_, and poly(EtEO_2_GE)_2.5k_. They are normalized per oxygen in polymer residue. The solid line shows $T_{\mathrm{CLP}}$ and the broken line shows $T_{\mathrm{CRP}}$.

**Figure 12 nanomaterials-13-01628-f012:**
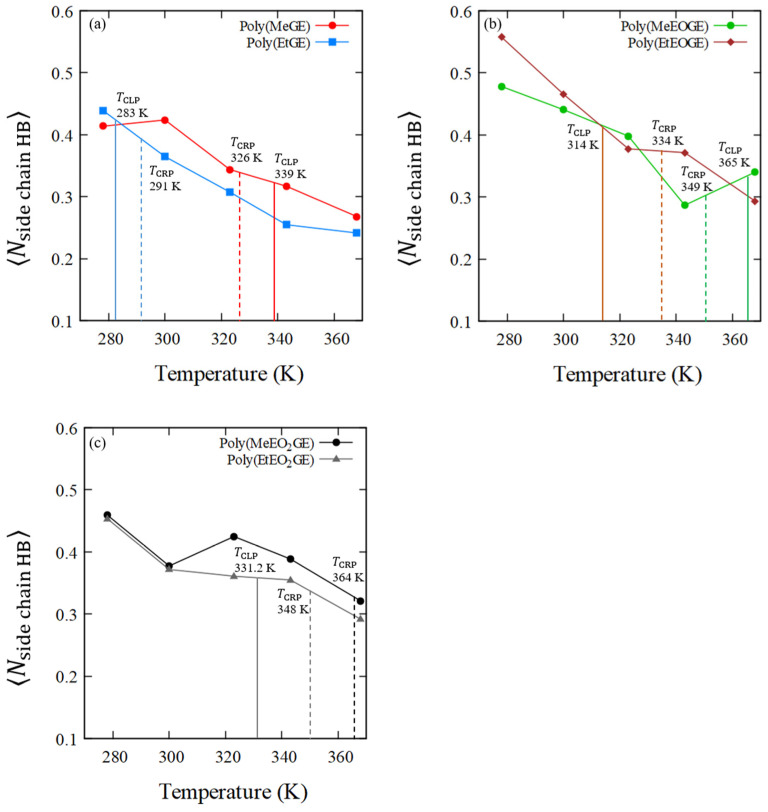
$\left\langle N_{\mathrm{side}\;\mathrm{chain}\;\mathrm{HB}}\right\rangle$ of (**a**) poly(MeGE)_2.5k_ and poly(EtGE)_2.5k_, (**b**) poly(MeEOGE)_2.5k_ and poly(EtEOGE)_2.5k_, and (**c**) poly(MeEO_2_GE)_2.5k_, and poly(EtEO_2_GE)_2.5k_. They are normalized per oxygen in polymer residue. The solid line shows $T_{\mathrm{CLP}}$ and the broken line shows $T_{\mathrm{CRP}}$.

**Table 1 nanomaterials-13-01628-t001:** Calculated principal *R_g_* peaks and experimental $T_{\mathrm{CLP}}$.

Sample ID	$\mathrm{Experimental}\;T_{CLP}*\;(K)$	*R*_g_/Å (SD/Å)				
		278 K	300 K	323 K	343 K	368 K
poly(MeGE)_2.5k_	338.5	8.6 (0.5)	8.9 (0.9)	8.7 (1.2)	8.6 (0.6)	9.1 (0.6)
poly(EtGE)_2.5k_	283.3	8.8 (0.3)	8.9 (0.5)	9.0 (0.5)	8.7 (0.5)	8.9 (0.5)
poly(MeEOGE)_2.5k_	364.6	9.7 (0.5)	9.4 (0.8)	9.3 (0.6)	9.0 (0.6)	9.5 (0.6)
poly(EtEOGE)_2.5k_	314.3	9.1 (0.5)	8.7 (0.4)	9.0 (0.5)	9.0 (0.6)	8.9 (0.5)
poly(MeEO_2_GE)_2.5k_	Not observed **	8.9 (0.2)	9.2 (0.4)	8.9 (0.4)	9.3 (0.4)	9.0 (0.5)
poly(EtEO_2_GE)_2.5k_	331.2	9.5 (0.2)	9.1 (0.5)	9.3 (0.5)	9.0 (0.5)	9.1 (0.6)

* Cloud point temperature observed by the turbidity measurements [10]. ** The LCST-type phase transition was not observed at temperatures up to 374 K [10].

**Table 2 nanomaterials-13-01628-t002:** Calculated $\bar{L}/L_{\max}$ at each temperature.

Polymer ID	Temperature (K)	$\bar{L}/L_{\max}$	$1/\sqrt{N}$	SD	CV
poly(MeGE)_2.5k_	278	0.14	0.19	0.04	0.32
	300	0.17	0.19	0.07	0.43
	323	0.21	0.19	0.07	0.34
	343	0.16	0.19	0.05	0.33
	368	0.15	0.19	0.07	0.44
poly(EtGE)_2.5k_	278	0.17	0.20	0.03	0.17
	300	0.20	0.20	0.06	0.31
	323	0.14	0.20	0.06	0.44
	343	0.15	0.20	0.05	0.35
	368	0.17	0.20	0.06	0.33
poly(MeEOGE)_2.5k_	278	0.28	0.23	0.10	0.37
	300	0.34	0.23	0.10	0.30
	323	0.25	0.23	0.10	0.39
	343	0.24	0.23	0.09	0.36
	368	0.25	0.23	0.10	0.39
poly(EtEOGE)_2.5k_	278	0.25	0.24	0.05	0.20
	300	0.23	0.24	0.08	0.35
	323	0.20	0.24	0.07	0.34
	343	0.23	0.24	0.08	0.35
	368	0.24	0.24	0.09	0.35
poly(MeEO_2_GE)_2.5k_	278	0.28	0.27	0.07	0.24
	300	0.27	0.27	0.09	0.33
	323	0.32	0.27	0.08	0.24
	343	0.27	0.27	0.11	0.40
	368	0.30	0.27	0.10	0.32
poly(EtEO_2_GE)_2.5k_	278	0.34	0.26	0.05	0.14
	300	0.30	0.26	0.11	0.37
	323	0.29	0.26	0.07	0.25
	343	0.28	0.26	0.11	0.38
	368	0.33	0.26	0.12	0.35

**Table 3 nanomaterials-13-01628-t003:** Calculated ${\left\langle \bar{L}\right\rangle }_{\mathrm{temp}}/L_{\max}$ with their SD and CV.

Polymer ID	${\left\langle \bar{L}\right\rangle }_{\mathrm{temp}}/L_{\max}$	SD	CV
poly(MeGE)_2.5k_	0.17	0.027	0.159
poly(EtGE)_2.5k_	0.16	0.023	0.140
poly(MeEOGE)_2.5k_	0.27	0.043	0.158
poly(EtEOGE)_2.5k_	0.23	0.020	0.085
poly(MeEO_2_GE)_2.5k_	0.29	0.024	0.082
poly(EtEO_2_GE)_2.5k_	0.31	0.026	0.084

**Table 4 nanomaterials-13-01628-t004:** Calculated $\bar{L}/L_{\max}$ of poly(MeEOGE) for molecular weights of 1250, 2500, and 5000.

Polymer ID	Temperature (K)	$\bar{L}/L_{\max}$	$1/\sqrt{N}$	SD	CV
poly(MeEOGE)_1.25k_	278	0.30	0.33	0.11	0.36
	300	0.43	0.33	0.05	0.12
	323	0.35	0.33	0.14	0.40
	343	0.37	0.33	0.12	0.32
	368	0.38	0.33	0.13	0.34
poly(MeEOGE)_2.5k_	278	0.28	0.23	0.10	0.37
	300	0.34	0.23	0.10	0.30
	323	0.25	0.23	0.10	0.39
	343	0.24	0.23	0.09	0.36
	368	0.25	0.23	0.10	0.39
poly(MeEOGE)_5k_	278	0.07	0.16	0.02	0.31
	300	0.11	0.16	0.01	0.12
	323	0.08	0.16	0.02	0.27
	343	0.15	0.16	0.03	0.20
	368	0.11	0.16	0.04	0.37

**Table 5 nanomaterials-13-01628-t005:** Time-averaged side chain lengths and their fluctuations.

Sample ID	Time-Averaged Side Chain Length (Å)					Temperature-Averaged Side Chain Length (Å)	SD (Å)	CV
	278 K	300 K	323 K	343 K	368 K			
poly(MeGE)_2.5k_	3.70	3.70	3.69	3.67	3.66	3.68	0.017	0.005
poly(EtGE)_2.5k_	4.74	4.72	4.70	4.69	4.68	4.71	0.023	0.005
poly(MeEOGE)_2.5k_	6.30	6.24	6.20	6.16	6.13	6.21	0.068	0.011
poly(EtEOGE)_2.5k_	7.06	7.01	7.08	7.11	7.00	7.05	0.047	0.007
poly(MeEO_2_GE)_2.5k_	8.08	8.00	8.08	8.01	7.97	8.03	0.048	0.006
poly(EtEO_2_GE)_2.5k_	8.54	8.54	8.49	8.56	8.55	8.54	0.024	0.003

**Table 6 nanomaterials-13-01628-t006:** Calculated $T_{\mathrm{CRP}}$ vs. experimental $T_{\mathrm{CLP}}$.

Sample ID	$\mathrm{Calculated}\;T_{\mathrm{CRP}}\;(K)$	$\mathrm{Experimental}\;T_{\mathrm{CLP}}\;(K)$	$T_{\mathrm{CRP}}$ $–T_{\mathrm{CLP}}\;(K)$
poly(MeGE)_2.5k_	326	339	−13
poly(EtGE)_2.5k_	291	283	+8
poly(MeEOGE)_2.5k_	349	365	−16
poly(EtEOGE)_2.5k_	334	314	+20
poly(MeEO_2_GE)_2.5k_	364	N.D. *	
poly(EtEO_2_GE)_2.5k_	348	331	+17

* N.D. means not detected.

**Table 7 nanomaterials-13-01628-t007:** Average numbers of hydrogen bonds between polymer and water calculated over 20 ns at 278, 300, 323, 343, and 368 K.

Sample ID	$T_{\mathrm{CLP}}\;(K)$	Average Number of Polymer–Water HBs in Main-Chain O Atoms, (Per Oxygen in Polymer Residue)					Average Number of Polymer–Water HBs in Side-Chain O Atoms, (Per Oxygen in Polymer Residue)				
		278 K	300 K	323 K	343 K	368 K	278 K	300 K	323 K	343 K	368 K
poly(MeGE)_2.5k_	338.5	0.31	0.29	0.25	0.23	0.19	0.41	0.42	0.34	0.32	0.27
poly(EtGE)_2.5k_	283.3	0.30	0.26	0.24	0.21	0.19	0.44	0.36	0.31	0.25	0.24
poly(MeEOGE)_2.5k_	364.6	0.34	0.31	0.27	0.20	0.23	0.48	0.44	0.40	0.29	0.34
poly(EtEOGE)_2.5k_	314.3	0.33	0.27	0.23	0.23	0.18	0.56	0.47	0.38	0.37	0.29
poly(MeEO_2_GE)_2.5k_	N.D.	0.39	0.27	0.25	0.28	0.21	0.46	0.38	0.42	0.39	0.32
poly(EtEO_2_GE)_2.5k_	331.2	0.35	0.29	0.28	0.26	0.22	0.45	0.37	0.36	0.35	0.29

## Data Availability

Data are available from the corresponding author upon request.

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
