# Peer review of "All-Atom Molecular Dynamics Simulations of the Temperature Response of Poly(glycidyl ether)s with Oligooxyethylene Side Chains Terminated with Alkyl Groups"

_nanomaterials, 2023, doi:10.3390/nano13101628_

Round 1

Reviewer 1 Report

My comments can be found in the attached file.

Author Response

We wish to express our strong appreciation for your insightful comments on our paper. We are thankful for the time and energy you expended. Our responses to your comments are in Word file.

Reviewer 2 Report

1. The authors use molecular dynamics simulations to study the thermal response of a series of polymers with alkyl-containing oligooxyethylene groups dissolved in water. The solvated polymers are introduced in a cubic simulation box along with the water molecules. The authors use a temperature ramp to heat their systems at the desired temperature where they perform molecular dynamics to observe the radius of gyration of the polymeric chains and the radial distribution of the water molecules along an indicative trajectory of 100 ns.   

2. In page 3, the authors repat the lines from 126 to 129.

3. The second peak of the bimodal distributions of the gyration radius shown in figure 2, correspond to conformations observed at the beginning of the simulations where the systems might not be well equilibrated.

4. The simulations of the small - weighted polymers run over 20 ns, in contrast to those of the lengthier polymers which run over 100 ns. The long-chained polymers obtain wider Rg distributions than those with small molecular weights. Is this because the small molecules are more compact or is due to the different time scales of the produced trajectories?

5. Molecular investigations of hydration shells around carbon nanoparticles and into their hydrophobic cavities has been also discussed. See https://doi.org/10.3390/computation7030050

6. By observing the conformational changes of the polymeric molecules during the simulated MD trajectories, the authors state that, the methyl - group polymers have more tendency to unfold into the aqueous solution than those with ethyl groups. Does that mean that the oxygen containing methyl groups are more hydrophilic than their ethyl- counterparts. 

7. How is the density of the hydration shells computed in the evaluations of the T_{CRP} in figures 6 and 7?

Author Response

Thank you very much for providing important comments. We are thankful for the time and energy you expended. Our responses to your comments are provided in the attached word file.

Reviewer 3 Report

The article can be reconsidered/published with significant improvement:

More citations were needed for better references such as the MM2 method.

How do the authors choose the length of the equilibration? It is necessary to confirm to the readers that the spatiotemporal scale of the equilibration is enough for the system with evidence such as the energy converging profile.

Is the forcefield temperature transferrable? Any evidence?

It would be good to include detailed forcefield parameters to easily facilitate the readers with information. 

No error bars for all the data?

Author Response

(The authors gave the same response as above.)

Round 2

Reviewer 2 Report

the manuscript is ready for publication in Nanomaterials

Author Response

We are glad to receive your comment.

Thank you for giving us the opportunity to strengthen our manuscript with your valuable comments and queries.

Reviewer 3 Report

Point 3 is not answered, which is vital. The author did not answer directly to "whether the forcefield is temperature transferrable" and did not show evidence. if the answer is no, then conclusion in this paper can not be supported at all. Furthermore, there are no detailed forcefield parameters provided in the paper.

Author Response

We apologize for the lack of explanation.

Our responses to your comments are provided in the attached word file.

Round 3

Reviewer 3 Report

This paper can be published in its present form, but I would suggest the forcefield parameters be included in the main manuscript as well to facilitate readers to reproduce easily.